# Mitigating Hallucinations in Large Language Models via Self-Refinement-Enhanced Knowledge Retrieval

Mengjia Niu
m.niu21@imperial.ac.uk
Imperial College London
United Kingdom

Hao Li
lihao350@huawei.com
Huawei
China

Jie Shi
SHI.JIE1@huawei.com
Huawei
Singapore

Hamed Haddadi
h.haddadi@imperial.ac.uk
Imperial College London
United Kingdom

Fan Mo
mofan1992@gmail.com
Huawei
China

## ABSTRACT

Large language models (LLMs) have demonstrated remarkable capabilities across various domains, although their susceptibility to hallucination poses significant challenges for their deployment in critical areas such as healthcare. To address this issue, retrieving relevant facts from knowledge graphs (KGs) is considered a promising method. Existing KG-augmented approaches tend to be resource-intensive, requiring multiple rounds of retrieval and verification for each factoid, which impedes their application in real-world scenarios.

In this study, we propose Self-Refinement-Enhanced Knowledge Graph Retrieval (Re-KGR) to augment the factuality of LLMs' responses with less retrieval efforts in the medical field. Our approach leverages the attribution of next-token predictive probability distributions across different tokens, and various model layers to primarily identify tokens with a high potential for hallucination, reducing verification rounds by refining knowledge triples associated with these tokens. Moreover, we rectify inaccurate content using retrieved knowledge in the post-processing stage, which improves the truthfulness of generated responses. Experimental results on a medical dataset demonstrate that our approach can enhance the factual capability of LLMs across various foundational models as evidenced by the highest scores on truthfulness.

## 1 INTRODUCTION

Large Language Models (LLMs) have demonstrated remarkable proficiency in generative tasks, where the question-answering (QA) system serves as a representative application [26]. However, despite the promising capabilities of LLMs, they are prone to "hallucinations", *i.e.,* generating responses not aligning with real-world facts to given inputs [16, 36]. This phenomenon impedes the wide deployment of LLMs and becomes particularly critical in high-risk scenarios, *e.g.,* healthcare, finance, and legal domains. Specifically in the healthcare area, nonfactual or incorrect information in medical

QA could lead to critical consequences for patients or even medical incidents [17, 25].

Hallucinations are inevitable for LLMs alone, due to the absence of domain-specific information or up-to-date knowledge in training datasets; or even presence, LLMs can struggle to well adapt to specialized domains or fail to memorize and apply these edge cases, especially when LLMs have small size (*e.g.,* 7B) and a low capacity [34]. To address this issue, a straightforward approach is integrating external knowledge into LLMs, which can be achieved through techniques such as Retrieval Augmented Generation (RAG) [17, 33]. RAG can resort to both unstructured documents [14] *e.g.,* papers, and structured database [17] *e.g.,* knowledge graphs (KGs). We employ well-structured KGs in this study since they can faithfully present domain-specific medical knowledge in the form of triples comprised of entities and corresponding relations [26], such as diseases and associated treatments, side effects, drugs and other complex mechanisms. Some research indicates that understanding structured knowledge in KGs can be hard for LLMs because the fundamental pattern that LLMs learn lies in semantic levels [38], not logistical levels. Nonetheless, there remains potential to directly inject the knowledge into LLMs' generations with a properly designed strategy to lower the bar for applying LLMs within the healthcare scenario [30].

Typically, RAG involves two key steps: i) retrieving relevant information from the data sources, and ii) fit this information into LLM generation. With such a design, *e.g.,* LLama-index, Langchain [1], the RAG approach has demonstrated its effectiveness in alleviating hallucinations, but they are not as efficient as we expected. Augmented generation is conventionally done by prompting LLMs to answer a question with the retrieved data, which still results in some level of hallucinations due to the LLM's limited reasoning capability. To further address this problem, recent work has explored post-processing techniques [17], leveraging retrieved knowledge to re-rank generated candidates or re-generate responses for better conversational reasoning. However, these methods incur substantial computational costs as they ask for external training for specific models or retrieve every factual statement present in the responses, including those that originate from the input questions and are generated with high confidence.

---

[1]https://www.langchain.com/

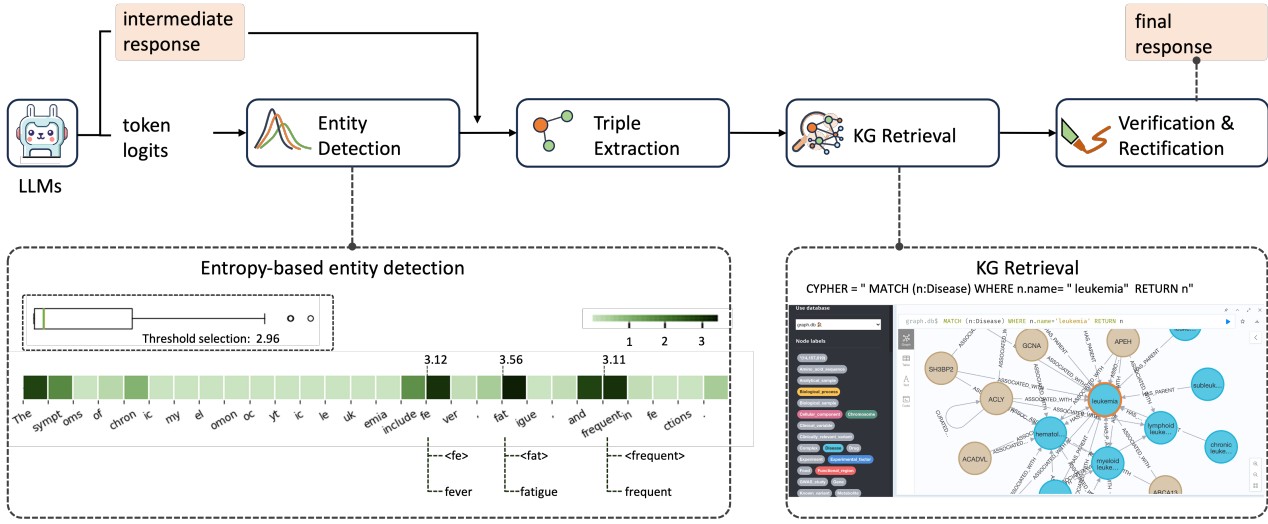

**Figure 1: Illustration of the proposed method, which consists of four components: (1) Entity Detection; (2) Triple Extraction; (3) Knowledge Retrieval; (4) Knowledge Verification & Rectification. Given an input question, the LLM generates an intermediate response along with relevant token logits, both of which are leveraged to identify critical word entities and subsequently extract a refined collection of factual statements, facilitating the KG retrieval and verification process. The lower-left subplot shows the mechanism of Entity Detection, which operates with the entropy of the next-token predictive probability distribution. This component dynamically identifies thresholds for anomaly detection on generated tokens for each sentence, contributing to the extraction of critical entities. The lower-right subplot visualizes a subgraph in the KG and presents an example query used for knowledge retrieval.**

To facilitate the redundant and cumbersome RAG steps, we propose Self-Refinement-Enhanced Knowledge Graph Retrieval (Re-KGR), which follows a refine-then-retrieve paradigm for injecting external knowledge in the post-generation stage with decreasing retrieval frequency to reduce hallucinations. Our work is inspired by recent studies examining the probability distributions associated with next-token prediction [10, 35] and analyzing the internal state of LLMs [3, 20]. These studies suggest that LLMs may know when they are prone to hallucinations. In such a way, we optimize the retrieval system by detecting segments, where unfaithful contents are more likely to occur, within LLMs' outputs, and refining the related factual statements requiring retrieval.

Figure 1 illustrates an overview of the proposed approach, which incorporates produced responses and pertinent token logits to reduce hallucinations. Specifically, Re-KGR method initially explores the attribution of next-token predictive probability distributions across different tokens and various model layers to identify word entities that are likely to be erroneous. Concurrently, from the generated text, we extract all factual statements as knowledge triples [26] but retain only those including the identified high-risk word entities, thereby reducing the number of retrieval instances. Subsequently, Re-KGR retrieves corresponding triples from a domain-specific KG and determines whether the triples in the refined set align with those from the KG. The original generated responses are then updated in accordance with the verification results.

We conduct experiments with a widely-used LLM, LLaMA [32], on a medical dataset (MedQuAD) along with a state-of-the-art

contrastive decoding technique (*i.e.,* DoLa [10]). Experiment results indicate that the proposed method can enhance the factuality of LLMs' responses with higher truthfulness scores.

In summary, the major contributions of this work are as follows:

- We introduce Self-Refinement-Enhanced Knowledge Graph Retrieval (Re-KGR) to mitigate hallucinations in LLMs' responses in the post-generation stage with minimal KG-based retrieval efforts.
- We implement Re-KGR method within a specific scenario, *i.e.,* medical question-answering tasks, and construct an expansive knowledge graph upon the existing corpus of medical information.
- Our experimental evaluation demonstrates the effectiveness of the proposed method, confirming great enhancements in improving response accuracy while concurrently achieving a notable reduction in time expenditure.

## 2 RELATED WORK

### 2.1 Hallucination in LLMs

Hallucination, which can refer to generated content that is nonsensical or deviates from the source knowledge [21], has attracted considerable research interest [31, 38]. Researchers suggest that hallucinations may stem from a lack of domain-specific information or up-to-date knowledge in training datasets [22, 41]. Additionally, it is posited that the intentional incorporation of "randomness" to

enhance the diversity of generated responses can also lead to hallucinations since it can increase the risk of producing unexpected and potentially erroneous content [19].

Previous studies [39] categorize hallucinations into three primary types: (1) input-conflicting, where outputs contradict the source content; (2) context-conflicting, which involves deviations from the previously generated context; (3) fact-conflicting, where outputs are against the established facts. In the medical domain, Ji *et al.* [18] have further refined these categories: *query inconsistency*, *fact inconsistency*, and *tangentiality*, which refers to responses that are topically related but do not directly address the posed question. After thorough analysis, they argue that fact inconsistency is the most common appearance of hallucination. In this study, we concentrate specifically on fact inconsistency hallucinations within medical QA tasks.

## 2.2 Hallucination Identification

Identifying where hallucinations derive in models is pivotal for enhancing the factuality performance of LLMs. Previous studies have highlighted that there are salient patterns in the model's internal representations associated with erroneous outputs during decoding. Research on the randomness of next-token predictive probability in the final layer has been a focus area. Lee *et al.* [19] propose that introducing "randomness" in the decoding process can encourage diversity but may result in factual inaccuracies. Xiao *et al.* [35] establish a connection between hallucinations and predictive uncertainty, quantified by the entropy of the token probability distributions, demonstrating that reducing uncertainty can decrease hallucinations. Similarly, Zhang *et al.* [37] calculate a hallucination score at the token level using the sum of the negative log probability and entropy.

Additional investigations suggest that the output distributions from all layers can be instrumental in detecting hallucinations. DoLa [10] suggests that divergence in distribution between the final layer and each intermediate layer may reveal critical entities requiring factual knowledge. Notably, significant divergence in higher layers indicates important entities. Chen *et al.* [7] attempt to use the sharpness of context activations within intermediate layers to signal hallucinations. These methodologies demonstrate significant efficiency in identifying hallucinations as they do not depend on comparing multiple responses or a specifically trained identification model.

## 2.3 Retrieval Augmented LLMs

Retrieving external knowledge as supplementary evidence to augment the truthfulness-generation capability of LLMs has emerged as a promising solution [23]. The effectiveness of such an approach is highly contingent upon the reliability of the external sources leveraged. Researchers have explored a variety of knowledge bases to acquire valuable assistive information, including public websites (*e.g.,* Wikipedia) [24, 40], web documents [14], open-source code data [8], and knowledge graphs [17, 28]. Nonetheless, not all obtained information is trustworthy, especially those derived from web content which may be uploaded without rigorous verification. Among the various knowledge resources, knowledge graphs stand

out since they contain high-quality expert knowledge [2], particularly preferable in the knowledge-intensive and high-risk medical domain.

In the implementation of LLMs, the retrieval-augmented module can be integrated at various stages of the generation process [31]. A large number of studies have investigated the conditioning of LLMs on retrieved information prior to the generation [27, 31], which still yields inaccurate responses due to LLM's limited reasoning processes. To address this issue and maximize the utility of additional info, some researchers have proposed to use it in the post-generation phase [11, 17, 28]. Dziri *et al.* [11] propose NPH, which trains a masked model to identify plausible sources of hallucination and resort to a KG for necessary attributions. RHO [17] gains KG embeddings with a TransE model and trains an autoencoder for producing multiple candidates, followed by a KG-based re-ranker for candidate response selection. KGR [15] retrieves all factual information in the responses and conducts a multi-round verification-generation process to obtain the final outputs. These approaches, while effective, often require supervised training or entail laborious retrieval efforts, which can hinder their application in a wide range of resource-intensive scenarios.

## 3 METHODOLOGY

In this section, we first outline our task in **Section 3.1**. Then, we introduce the proposed Self-Refinement-Enhanced Knowledge Graph Retrieval (Re-KGR) for hallucination mitigation. The core idea of our approach is to verify and rectify factual information contained in generated content efficiently by identifying segments that have a high likelihood of inaccuracy in advance. As illustrated in Figure 1, Re-KGR incorporates both generated texts and associated token logits to conduct the mitigation task in the post-generation stage. Specifically, in **Section 3.2**, we describe how to identify critical entities, namely those with a higher risk of hallucination. Subsequently, we delineate how to extract and refine factual statements in the form of factual triples based on the generated texts and identified critical entities in **Section 3.3**. These triples are then employed for KG retrieval as shown in **Section 3.4**. Finally, in **Section 3.5**, we detail the process of verifying the refined knowledge statements against KGs to produce factually aligned responses.

## 3.1 Problem Formulation

Given a sequence of tokens $\{x_1, x_2, \ldots, x_{s-1}\}$, LLMs can predict the next token $x_s$, which is decoded from the next-token distribution $p(x_s|x_{<s})$, denoted as $p(x_s)$ for short. In the context of a QA task, LLMs are able to formulate a reasonable answer $\mathcal{A}$. The generated answer can be deconstructed into an aggregation of simple factual statements, typically structured as knowledge triples `t = <[SBJ], [PRE], [OBJ]>`, where `[SBJ]` and `[OBJ]` denote the subject and object entities respectively, and `[PRE]` is the relational predicate [17]. For example, `<Adult Acute Lymphoblastic Leukemia, diagnosed by, blood tests>`. Healthcare QA represents a domain-specific instance of the QA task, where the factual set for each given question is expected to be aligned with established expert medical knowledge. We present the collection of critical factual statements as $T = \{t_i\}_{i \in \{0,1,\ldots,I\}}$, where $I$ is the number of triples. The objective of our research is to efficiently extract the

set $T$ from generated $\mathcal{A}$ and verify the truthfulness of necessary $t_i$ against a reliable knowledge repository. Once verified, the newly obtained triples allow us to refine the original answer to ensure it is accurate and reliable.

## 3.2 Entity Detection

Entity Detection aims to identify word entities that are likely to be inaccurate in advance, thereby relieving the demands of the subsequent verification process. This detection mechanism is based on analyzing the predictive uncertainties of token logits. Xiao *et al.* [35] suggest that a higher predictive uncertainty, as quantified by the entropy of token probability distributions, is indicative of an increased propensity of hallucinations. Apart from outputs of the final layer, some researchers have focused on the connection between the inner representations of intermediate layers and hallucinations, such as DoLa [10] utilizing output distribution divergence among different model layers and the in-context sharpness alert method developed by Chen *et al.* [7].

Inspired by the aforementioned cases, our approach quantifies the predictive uncertainty of the next token from the following insights: i) Considering the next-token distribution $p(x_s)$, we examine the uncertainty through the max value $c_m = max(p(x_s))$ and the entropy $c_e$, which is denoted as follows:

$$c_e = \mathcal{E}(p(x_s))$$
$$\mathcal{E} = -\Sigma_{d \in \{1,2,\cdots,D\}} p(x_s^d) \log p(x_s^d),$$

where $\mathcal{E}(\cdot)$ denotes the entropy function and $D$ is the length of token vector. ii) As for the distribution divergence between token logits $q(x_s|x_{<s})$ ($q(x_s)$ for short) at the final layer $N$ and each intermediate layer $j \in J$, we refer to observations from *DoLa* [10]. When predicting important predictions requiring knowledge information, the divergence would be still high in the higher layers. Thus, we measure the uncertainty $c_{js}$ as:

$$c_{js} = max \left( \mathcal{D}(q^N(x_s), q^j(x_s)), \ j \in J^* \right) \quad (1)$$

where $J^*$ is the integration of candidate intermediate layers and $\mathcal{D}(\cdot)$ presents the probability distribution distance between layer $N$ and $j$, separately. The distance is measured by Jensen-Shannon divergence. Upon obtaining the criteria for each token within a response, we detect abnormal values for each uncertain criterion individually through quartile-based assessment. Subsequently, the associated tokens can be identified. An example of the whole process is presented in the lower left subplot of Figure 1 and an empirical evaluation of these criteria is elaborated in Section 5.

## 3.3 Triple Extraction

Triple Extraction aims to identify knowledge components that are likely to be inaccurate. A coherent response from LLMs typically comprises a series of factual statements that may characterize a specific entity or delineate the relationship between two entities [12]. These statements are structurally presented as triples `t = <[SBJ], [PRE], [OBJ]>`. To accomplish this, we start with a prompted LLM to enumerate all triples in a given response. Afterwards, as depicted in Figure 2, we exploit extracted triples along with identified uncertain entities to derive the final triple set $T_f = \{t_i^0\}_{i \in \{1,2,\ldots,I\}}$ (*i.e., $T$* mentioned in Section 3.1). This refined set

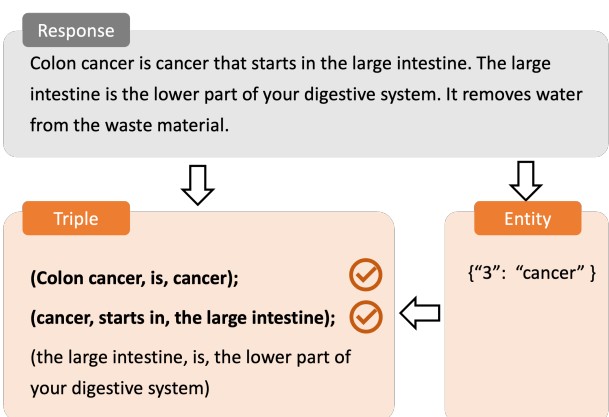

**Figure 2: Example of Triple Extraction Process. LLMs are prompted to extract factually related statements in the form of knowledge triples, with identified entities being utilized to select critical triples.**

simplifies the KG verification procedure by removing redundant information, thus enhancing the overall efficiency.

## 3.4 KG Retrieval

After assembling the collection of factual triples, the retrieval module aims to resort to an external knowledge base to obtain associated factual information. For this purpose, we select KGs as our preferred external base since they are considered solid sources containing sufficient expert knowledge. As shown in Section 3.1 and 3.3, such knowledge is encapsulated faithfully in the form of semantic triples. An example of how to query from KG and a subgraph is presented in the low right in Figure 1.

Previous research on KG retrieval has primarily relied on supervised fine-tuned models such as TransE [5, 17], or intricately crafted systems [12, 15] without disclosing detailed retrieval mechanisms. Our approach prioritizes a straightforward strategy focusing on synonym detection and similarity assessment. To elaborate, we first expand the original triple set $T_f$ into an enriched set $T_e = \{t_i^{e_i}\}_{i \in \{1,2,\ldots,I\}, \ e_i \in \{1,\ldots,E_i\}, \ E_i \in \mathcal{N}}$ by searching and adding synonyms for each entity and predicate. Then utilizing $T_f$ and the expanded set $T_e$, we retrieve associated knowledge triples from a comprehensive medical KG. The retrieved triple set is denoted as $T_g = \{t_i^{g_i}\}_{i \in \{1,2,\ldots,I\}, \ g_i \in \{1,\ldots,G_i\}, \ G_i \in \mathcal{N}}$, where $G_i$ is the number of all retrieved triples for each $t_i^0$. This enriched set will be utilized for the following verification.

## 3.5 Knowledge Verification & Rectification

Knowledge Verification aims to confirm the truthfulness of the selected triples. In this study, we employ semantic-similarity measurement models $\mathcal{S}(\cdot)^2$ to evaluate the consistency between $t_i^0$ and the corresponding retrieved $\{t_i^{g_i}\}$. Similarity score vector is defined

---

[2] one can use the *spaCy* package from https://github.com/explosion/spaCy

as $\mathbf{s}_i = \mathcal{S}(t_i^0, \{t_i^{g_i}\})$. The final triple is illustrated as:

$$t_i' = \begin{cases} t_i^0, & \text{if } max(\mathbf{s}_i) > \tau, \\ t_i^g, \ g = \underset{g_i}{\arg\max}\,(\mathbf{s}_i) & \text{if } max(\mathbf{s}_i) \leq \tau \end{cases} \quad (2)$$

where $\tau$ is a predefined threshold. In the final Rectification stage, newly incorporated triples are harnessed to substitute the corresponding content, while a grammar-correction model is utilized in the end to guarantee the integrity of the final output.

## 4 EXPERIMENTS

### 4.1 Experimental Setup

**Dataset** *MedQuAD* [4, 18] dataset includes real-world medical QA pairs that have been compiled from various National Institutes of Health websites. This comprehensive collection encompasses 37 different types of questions, addressing a wide range of topics including treatments, diagnoses, and side effects. Moreover, it covers a variety of medical subjects such as diseases, medications, and other medical entities like tests. During experiments, we eliminate certain repetitive QA pairs and additionally filter selected QA pairs by focusing on those where disease names are present within the built KG. However, it is necessary to acknowledge that there remain QA pairs that contain entity nodes but are not associated with a fully annotated KG triple.

**Baselines** We select LLaMA-7b [32] and its contrastive-decoding version, *i.e.,* DoLa [10] as the foundational models to evaluate the effectiveness of our Re-KGR approach on hallucination mitigation. The LLaMA-7b model is pre-trained on trillions of tokens and provides a strong foundation for research in the field of natural language processing. DoLa, specifically tailored for LLaMA model, introduces a novel contrastive-decoding method. Typically, it is designed to reduce hallucination by contrasting the output probability distributions derived from the final layer and various intermediate layers within the model. The vanilla LLaMA-7b and DoLa also serve as baselines in our research. We refer to existing repositories [3] for the deployment.

### 4.2 Implementation

Following the deployment of DoLa, we utilize *transformers 4.28.1* [4] for all experiments. The LLaMA-7b is also downloaded from it. In the KG Verification stage, we utilize a domain-specific knowledge graph to ensure retrieval accuracy. We employ the Clinical Knowledge Graph (CKG) [29][5], which integrates knowledge from multiple widely-used biomedical databases, as our foundational graph database. Considering that the CKG lacks certain essential information, such as disease symptoms, we supplement it with the PrimeKG [6], a multimodal knowledge graph specifically designed by Chandak *et al.* for precision medicine analyses. This integration enhances the comprehensiveness of the utilized knowledge graph. To align with the requirements of the CKG, we leverage the *Neo4j 4.2.3* [6], a graph database management system as shown in the lower right in Figure 1, to store and query knowledge triples.

---

[3]https://github.com/voidism/DoLa/tree/main
[4]https://huggingface.co/docs/transformers/v4.28.1/en
[5]https://ckg.readthedocs.io/en/latest/INTRO.html
[6]https://neo4j.com/

**Table 1: Main results on MedQuAD based on the GPT-4 judgements.**

| Model | GPT-4 scores | Perf. ↑ |
|---|---|---|
| LLaMA-7b [32] | 0.527 | - |
| **+ Re-KGR** | 0.537 | 1.90% |
| DoLa [10] | 0.591 | 12.14% |
| **+ Re-KGR** | **0.610** | 15.75% |

### 4.3 Evaluation Protocols

GPT-4 [1] has been empirically validated for its capability to automatically evaluate the quality of generated responses [9]. Referring to studies from Chiang *et al.* [9] and Feng *et al.* [13], we employ GPT-4 as an examiner to evaluate the truthfulness of the responses. Specifically, for each question, we utilize meticulously designed prompts to instruct GPT-4 to assess each response against the given standard answer and its extensive knowledge base. The evaluation score ranges from 0 to 1. Here, 0 indicates that the generated answer is either irrelevant or factually incorrect, while 1 represents perfect alignment with the ground-truth information.

## 5 RESULTS AND ANALYSIS

### 5.1 Overall Performance

Table 1 presents the experimental results of the automatic evaluation of GPT-4 on the MedQuAD dataset. The results demonstrate that our Re-KGR method surpasses the baselines, yielding significant enhancements in generating factually accurate responses for medical QA tasks of LLMs. Typically, when implemented alongside the DoLa model, our method achieves the highest level of truthfulness performance with a score of 0.610, indicating a 15.75% improvement over the vanilla LLaMA and a 3.21% improvement compared to DoLa. Nonetheless, when integrated into vanilla LLaMA, the improvement is a modest 1.90%, and the truthfulness score is lower than that achieved with the original DoLa approach. This underperformance could be attributed to the intrinsic limitations of models with different decoding methods. For example, vanilla LLaMA-7b has a tendency to generate brief, repetitive, or irrelevant content. This reduces the need for verification, as it tends to overlook repetitive hallucinations or irrelevant content, regardless of the sufficient knowledge contained in the KG. The augmented performance is limited as well.

### 5.2 Additional Studies

In this section, we further present the performance of various criteria for entity detection and assess the effectiveness of the refine-then-retrieve paradigm.

The $c_{js}$-**based criterion is more efficient for critical entity detection while the** $c_e$-**based criterion demonstrates superior performance on the original hallucination mitigation task with a higher evaluation score.** We conducted additional experiments on the *MedQuAD* dataset to investigate the performance of various criteria. Table 2 presents the automatic GPT-4 evaluation results and Figure 3 delineates the consequence of critical entity and triple identification. It is noteworthy that the $c_{js}$-based

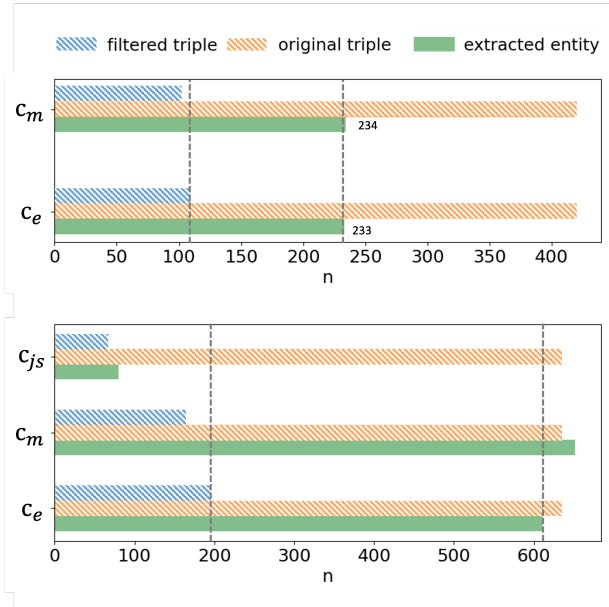

**Figure 3: Illustration of entity detection results across various criteria and corresponding outcomes of self-refinement triples. The upper figure displays the number of extracted entities, refined triples, and original triples for Re-KGR method using the LLaMA-7b model, while the lower one presents results for the method with the DoLa model. Grey dashed lines indicate the number of extracted entities and refined triples based on the entropy criterion $c_e$, respectively.**

**Table 2: Evaluation results on the effectiveness of different criteria, namely the max output distribution divergence between the final layer and intermediate layers ($c_{js}$), the max value ($c_m$) and the entropy ($c_e$) of the next-token prediction probability. The GPT-4 scores for answers obtained with various criteria are utilized for evaluation.**

| Model | GPT-4 scores | | |
|---|---|---|---|
| | $c_{js}$ | $c_m$ | $c_e$ |
| DoLa [10]+Re-KGR | 0.604 | 0.603 | 0.610 |

method extracts fewer entities and greatly reduces the number of triples requiring verification without a significant compromise on performance, indicating its effectiveness in enhancing the quality of information retrieval. Moreover, as depicted in Figure 3, this method maintains a higher ratio of selected triples to detected entities in comparison to the selection strategies based on the $c_m$ and $c_e$ scores. However, the $c_{js}$-based method is specifically tailored for the DoLa model, since it requires calculating the JS divergence between output distributions from relevant layers for the purpose of contrastive decoding.

While $c_{js}$-based method is not directly applicable to most open-source LLMs, $c_m$ and $c_e$ are readily available. As illustrated in Table

**Table 3: Comparative analysis of the average retrieval time and overall scores on truthfulness, as assessed by GPT-4, for various approaches under different triple refinement conditions.**

| Model | KG retrieval time (s) | GPT-4 score |
|---|---|---|
| LLaMA + Re-KGR | 16.14 | 0.537 |
| - without refinement | 64.88 | 0.518 |
| DoLa + Re-KGR | 26.13 | 0.610 |
| - $c_{js}$-based | 14.13 | 0.604 |
| - without refinement | 71.29 | 0.600 |

2, the $c_e$-based method achieves the best performance with the highest evaluation score. Given that the $c_e$-based method necessitates more retrieval attempts as presented in Figure 3, both $c_{js}$ and $c_m$-based approaches may overlook some necessary triples. Figure 3 shows that Re-KGR achieves a higher ratio of filtered to original triples on vanilla LLaMA compared to DoLa, while the latter indicates better performance. Such observation suggests that contrastive decoding by DoLa may direct the predictive distribution towards a direction where uncertainty is expressed more truthfully. This helps to identify and verify a greater number of critical triples, effectively enhancing the truthfulness of LLMs.

**The self-refinement process in triple extraction reduces the time required for retrieval without compromising the augmented impact of KGs on LLMs.** Table 3 displays the average retrieval time per question and the overall scores on truthfulness evaluated by GPT-4 for the proposed method across various foundational models. Compared to retrieval without triple refinement, Re-KGR method reduces retrieval time by 63% to 75%. The implementation of the $c_{js}$-based method can achieve additional reductions in retrieval time. In contrast to the decrease in retrieval time, we can see an increase in the evaluation scores of the proposed method. This result, *i.e., retrieving all triples results in worse performance*, may caused by verification and rectification of unnecessary triples where the semantic meaning may be incorrect. For example, in querying the symptoms of breast cancer, both <the symptoms of breast cancar, is, XXX> and <breast cancer, is, XXX> could be extracted from the same response, but the latter may lead to an error revision on the output. By triple refinement, we can refine the triple set with critical entity [symptoms].

In summary, by employing critical entities to select pivotal triples before conducting KG retrieval, our method substantially reduces the required retrieval time without diminishing the augmentative impact of retrieval on LLMs. We also argue that integrating the contrastive decoding module, as in DoLa, and leveraging the $c_{js}$-based criterion for identifying necessary entities and triples will further facilitate the whole truthfulness response generation process.

## 6 CONCLUSION AND FUTURE WORK

In this paper, we introduce Self-Refinement-Enhanced Knowledge Graph Retrieval to efficiently mitigate hallucinations in LLMs' responses by incorporating structured KGs and minimizing retrieval

efforts on medical QA tasks. Our Re-KGR resort to external knowledge in the post-generation stage considering that the direct injection of such knowledge as prompts to LLMs can still lead to some levels of hallucination due to the LLM's limited reasoning processes. Additionally, we leverage properties of next-token predictive probability distributions across different tokens and various model layers to identify tokens with a high likelihood of hallucination in advance and refine the collected knowledge triple set, reducing subsequent retrieval costs. Experimental results on the MedQuAD dataset demonstrate that our approach achieves greater performance on hallucination mitigation via a higher score on truthfulness assessed by GPT-4 and ground-truth answers. This underscores that Re-KGR can enhance the factual capability of LLMs across various foundational models. Further studies have been conducted to explore the underlying mechanism of our method, including the evaluation of various criteria for entity detection and the assessment of the effectiveness of the self-refinement module on triples.

Not requiring specific training processes, our Re-KGR can be readily applied to various downstream tasks provided there are well-constructed domain-specific knowledge graphs, though it is primarily targeted at the medical field. In the future, we will investigate the generalization capability of the proposed method across various scenarios given an applicable knowledge graph. Additionally, it would be beneficial to explore the integration of retrieval processes during the generation phase to further cut down overall generation time.

## ACKNOWLEDGMENTS

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
