# OpenReview forum: "Mitigating Hallucinations in Large Language Models via Self-Refinement-Enhanced Knowledge Retrieval"
_ACM.org/SIGIR/2024/Workshop/Gen-IR — Gen-IR_SIGIR24_

### Official Review · Reviewer_srjt · 2024-05-22
**This paper introduces a novel approach called Re-KGR, which aims to reducing the hallucination senarios occured in LLLMs.**

**Rating:** 1
**Confidence:** 3

**Review:**

This paper introduces a novel approach called Re-KGR, which aims to reduce the hallucination scenarios that occur in LLMs.  By detecting the entities that are prone to cause hallucination, and then extracting the triplet knowledge triples, i.e. the subject, predicate, and object, the proposed method uses the KG retrieval to retrieve relevant knowledge triples and finally compare the generated triples with the retrieved triples and rectify the inconsistency.

- Strength:
Overall, this paper presents very well and the proposed method is novel. It also applies the proposed method to the recent popular Llama model and its variant model: DoLa. The experiment results on MedQuAD show that the proposed method can lead to the performance improvement to some extent.

- Weakness:
1. although this paper has implemented the Re-KGR method on two LLMs, but the difference between LLaMA and DoLa is only their decoding strategy. In Table 1, it seems Re-KGR is more effective on the LLM with contrastive decoding than on the LLaMA model, I would be more curious about its generalization across different decoding methods/models.
2. In addition, the baseline models in this paper is only the model prior to Re-KGR implementation models, the authors could compare to more baseline models to analyze the effectiveness of the proposed method.
3. there are some typos: on page 3. line 343, there are duplicate "to".

---

### Official Review · Reviewer_UUYn · 2024-05-28
**An adaptation of Dola for the medical domain, with some additional sensitivity studies related to the task and some novel ideas for entity detection.**

**Rating:** 1
**Confidence:** 4

**Review:**

## Strengths:

- the authors drill down into what is their current issue they address in a very clear way and end up with 'fact inconsistency hallucinations within medical QA tasks'.
- the literature review is thorough and well structured into hallucination definition, identification and mitigation.
- the authors examined several aspects of the model in detail, such as the entity extraction and the retireval time.

## Weaknesses:

- The authors did not benchmark other RAG baselines. Some simple baselines, like feeding knowledge to the prompt via dense retrieval, might deserve to be there. I also understand that such a simple basline does require some engineering work and that there might not be an available codebase for this specific case of fact verification.
- Figure 1's examples in dashed boxes seem to relate leukemia with breast cancer. Is that on purpose? If yes, I did not understand the logic behind it.

## Minor comments

- Suggestion: Maybe the title of the paper could have the word 'Graph' in it to make it very obvious to the reader.
- Line 89:  'argument this information into LLM generation' -> 'fit this information into LLM generation' ?
- Line 106 & 573: ' refine-then-retrieval' -> ' refine-then-retrieve'
- Line 229: 'are against from the established facts' -> 'are against the established facts'
- Line 274: '(e.g., the Wikipedia)' -> '(e.g. Wikipedia)'

---

### Official Review · Reviewer_bJWL · 2024-05-28
**Alleviating hallucinations in LLMs by incorporating knowledge graph retrieval with decreased retrieval frequency**

**Rating:** 1
**Confidence:** 4

**Review:**

This paper proposes alleviating the issues of hallucinations in LLMs by incorporating knowledge graph retrieval with decreased retrieval frequency. The experiments are conducted on one medical dataset, and the results show the effectiveness and efficiency of the proposed method.

Strengths:

1. The paper is well-written and easy to understand.
2. The motivation to augment the trustworthiness of LLMs' generation is an important and timely topic.
3. The experimental settings are reasonable.

Weaknesses:

1. For triple extraction, I wonder whether the authors have considered the semantic similarity between the detected uncertain entities and the extracted triples, instead of exact matching.
2. The verification process could be further improved. The current method relies solely on the similarity between the detected triple and the retrieved triple. How can we ensure the retrieval accuracy?
3. There is a performance difference between LLaMA and DoLa. It would be beneficial to discuss the potential reasons for this discrepancy.

---

### Official Review · Reviewer_nutd · 2024-05-30
**Knowledge graph based retrieval for token verification**

**Rating:** -1
**Confidence:** 4

**Review:**

This paper focuses on alleviating hallucination during large language model generation. This paper claims that tokens with high uncertainty are related to hallucinated output, thus the authors propose to extract entities based on the token probability distribution during the next token prediction. The extracted entities are utilized for retrieval on the knowledge graph. By retrieval, the uncertain entities are replaced or enhanced by newly-retrieved entities, then the hallucination would be alleviated. I have several concerns.
1. The improvement is limited, for about only 2-3%
2. No other hallucination control baselines are compared, so it is hard to verify the superiority of the proposed method
3. This method is similar to retrieval augmented generation, but no comparison is done between these two. The novelty is limited.

---

### Decision · Program_Chairs · 2024-05-28

**Decision:**

Accept

**Comment:**

This paper proposes an approach for reducing hallucinations using KG triples to check LLM generations. Reviewers liked the paper, but noted that the baselines are limited. The authors could consider strengthening the paper by addressing this point.